# Image-Acceleration Multimodal Danger Detection Model on Mobile Phone for Phone Addicts

**DOI:** 10.3390/s24144654

**Published:** 2024-07-18

**Authors:** Han Wang, Xiang Ji, Lei Jin, Yujiao Ji, Guangcheng Wang

**Affiliations:** School of Transportation and Civil Engineering, Nantong University, Nantong 226019, China; hanwang@ntu.edu.cn (H.W.); 2233320017@stmail.ntu.edu.cn (X.J.); 2033110237@stmail.ntu.edu.cn (L.J.); 2333320005@stmail.ntu.edu.cn (Y.J.)

**Keywords:** multimodal danger detection model, phone addicts, mobile phone, rear camera, gravitational acceleration sensor

## Abstract

With the popularity of smartphones, a large number of “phubbers” have emerged who are engrossed in their phones regardless of the situation. In response to the potential dangers that phubbers face while traveling, this paper proposes a multimodal danger perception network model and early warning system for phubbers, designed for mobile devices. This proposed model consists of surrounding environment feature extraction, user behavior feature extraction, and multimodal feature fusion and recognition modules. The environmental feature module utilizes MobileNet as the backbone network to extract environmental description features from the rear-view image of the mobile phone. The behavior feature module uses acceleration time series as observation data, maps the acceleration observation data to a two-dimensional image space through GADFs (Gramian Angular Difference Fields), and extracts behavior description features through MobileNet, while utilizing statistical feature vectors to enhance the representation capability of behavioral features. Finally, in the recognition module, the environmental and behavioral characteristics are fused to output the type of hazardous state. Experiments indicate that the accuracy of the proposed model surpasses existing methods, and it possesses the advantages of compact model size (28.36 Mb) and fast execution speed (0.08 s), making it more suitable for deployment on mobile devices. Moreover, the developed image-acceleration multimodal phubber hazard recognition network combines the behavior of mobile phone users with surrounding environmental information, effectively identifying potential hazards for phubbers.

## 1. Introduction

With the rapid improvement in economic levels and the swift development of intelligent devices, smartphones are gradually becoming an indispensable part of people’s daily lives. Smartphones have brought convenience to people while also triggering a series of social issues. In particular, smartphones are increasingly encroaching on people’s fragmented time, with more and more individuals becoming addicted to mobile entertainment activities [1]. The advent of smartphones in the 21st century, with their powerful features and portability, has given rise to “phubbers”—a group of people who are constantly using their phones, regardless of time or place. This group is growing rapidly and is visible everywhere [2]. The continuous habit of looking down at smartphones poses a serious threat to both individual health and public safety. On one hand, prolonged smartphone use can lead to visual fatigue, numbness in the wrists, and physical ailments such as cervical spondylosis and mental distraction [3]. The accumulation of these discomforts can easily escalate into health risks. On the other hand, when “phubbers” are out and about, they often disregard potential dangers in their surroundings, which can lead to a series of accidents. For instance, they may fall down stairs or fail to yield to oncoming vehicles when crossing the road. These accidents not only harm individuals but also pose a threat to the safety of others, even endangering lives [4]. Therefore, research on detecting and warning against hazardous states for “phubbers” is of significant importance for ensuring individual travel safety and maintaining traffic order.

Currently, there are various methods for detecting and warning against hazardous states for “phubbers”. Based on the type of sensors used and the implementation methods, the main types can be categorized into monitoring device detection methods, wearable device detection methods, and mobile phone detection methods. The detection method based on monitoring devices usually uses cameras to detect and warn about the behavior of people playing with mobile phones in specific areas, such as subway station staircase areas and vehicle cabins. The detection methods based on monitoring devices usually use cameras to detect and warn individuals about playing with mobile phones in specific areas, such as intersections and driver’s cabins [5]. Due to the fixed and immovable monitoring equipment, the regulatory scope of these methods is limited. The detection method based on wearable devices usually monitors and gives alerts about the user’s behavior of playing with their phone while looking down through sensors on their body [6]. These methods can achieve real-time monitoring of user behavior without being limited by time or scenario. However, users need to bear the cost of purchasing the device, and long-term wearing of some wearable devices can cause certain discomfort. The detection methods based on mobile phones usually assume that the mobile phone user is currently using the phone, and directly uses the phone camera to recognize the user’s current environment, and provides alerts based on the environmental recognition results [7]. However, due to the lack of control over the timing of camera activation, it is easy to miss the warning time, and there is rapid energy consumption generated by keeping the camera on.

Based on the above analysis, it can be concluded that the existing methods for detecting and warning about the behavior of individuals with their heads down have the following shortcomings: The existing methods use a single sensor to collect observation data and study the recognition model of a single task, such as using the phone or scene recognition. Due to the lack of research on multi-source data models that combine user behavior analysis with surrounding environment recognition, it is impossible to achieve adaptive energy-saving control of mobile phone cameras and high accuracy in identifying user danger states.

In response to the above issues, this article proposes a multimodal smartphone addiction danger perception and warning system based on mobile phones. The system combines user behavior analysis with scene recognition, and reduces battery loss on the mobile phone by controlling the opening of the camera based on the results of behavior analysis. Subsequently, a lightweight multimodal “phubbing” danger perception model, based on MobileNet, is constructed to effectively integrate the user motion state and surrounding environmental information, accurately identifying potential “phubbing” danger states. Finally, the optimized model is deployed in the Android environment to achieve high accuracy and real-time perception and warning of danger states. The system framework is illustrated in Figure 1.

The main contributions of this paper can be summarized as follows:A mobile-end rear-view image-gravity acceleration multimodal “phubbing” danger state monitoring dataset was constructed, and the behaviors of smartphone users were combined with surrounding environmental information, leading to the proposal of a lightweight image-acceleration multimodal “phubbing” danger perception network model for mobile devices.The proposed multimodal lightweight network model has been successfully deployed on Android devices. Online experiments have demonstrated that the model achieves high accuracy in identification and operates at a fast speed, showcasing promising practical applications and potential for wider adoption.User behavior analysis results based on acceleration data are used to control the turning on of mobile phone cameras, thereby reducing the battery consumption of the phone.

## 2. Related Works

In the field of monitoring devices, Shi et al. [8] proposed a method for recognizing “phubbing” behavior while driving based on a hybrid dual-stream convolutional neural network. The results show that with adoption of the method, the accuracy values for face recognition and fatigued driving detection are 1.36 and 2.5 percentage points higher, respectively, than with other methods. Goh et al. [9] installed BLE beacons in commonly hazardous areas such as sidewalks, stairs, and crosswalks to provide relevant alerts to users. A field study with 24 participants validated the effectiveness of Smombie Forecaster (smartphone pause time increased by 1.59 times; average frequency of steps decreased from 1.68 Hz to 1.47 Hz).

For wearable device-based approaches, Jia et al. [6] proposed a model for evaluating neck muscle fatigue based on amplitude–frequency joint analysis. This model utilizes surface electromyography (sEMG) signals collected by sEMG-JASA over different time intervals to recognize users’ neck muscle activity, thereby achieving detection of “phubbing” behavior and fatigue warnings. The model can alert users to fatigue, help users form good mobile phone usage habits, and provide a basis for the development of an intelligent wearable system. Bi et al. [10] proposed the CSEar system, which utilizes built-in accelerometers in off-the-shelf wireless earphones. Through the metasensing model, it detects various head poses and gait signals of users in resting and walking states, thus identifying whether users are engaged in smartphone-related activities; the maximum accuracy can reach 93%. Haobo et al. [11] employed Bi-LSTM to extract features from continuous activity streams captured by three inertial sensors worn on the user’s wrist, waist, and ankle. This process aims to identify whether “phubbers” are at risk of potential falls. The proposed hybrid approach is shown to yield an average classification accuracy of approximately 96% while improving performance and robustness across all participants.

In the field of mobile phones, Donghee et al. [12] designed a smartphone application called “Smombie Guardian”. This application utilizes target detection algorithms to detect obstacles while “phubbers” are walking and issues warnings to prevent potential collision accidents. The results of our field test study involving 74 human subjects demonstrated the feasibility of the app with respect to its effectiveness, usefulness, and unobtrusiveness. Hyun-Seok et al. [13] proposed a deep learning method based on smartphone target detection. Using MobileNet, it detects stairs and pedestrian crossings, providing alerts for imminent risks. The proposed solution facilitates detecting and circumventing accident-prone situations for users. The smartphone being used detects dangers directly and informs the user via the screen. Chang et al. [14] developed a mobile application called “phubber Minder” [15]. It utilizes semantic segmentation to identify the boundaries of sidewalks and lanes, issuing alerts via smartphones before “phubbers” enter the lanes. Experiments prove that P-Minder achieves 75.52% detection accuracy when used in practice, and it can run smoothly on the test mobile phone.

## 3. Multimodal Phubbing Danger Detection Dataset

Due to the absence of publicly available multimodal datasets matching rear-view images with gravitational acceleration for monitoring the risk of “phubber” behaviors, we constructed our own dataset for head-down risk detection based on image-gravitational acceleration data. The specific collection and production process is as follows: Firstly, we defined common hazardous scenarios for “phubbers” when they are outdoors, including in stair areas, pedestrian crossing areas, areas with surface water accumulation, and low-light areas. At the same time, we defined common mobile phone user behavior: using the phone in a stationary state, walking on flat ground while using the phone, and walking upstairs or downstairs while using the phone. We then combined the defined hazardous scenarios with mobile phone user movement states to create nine common “phubber” risk states (Figure 2), including walking upstairs or downstairs while using the phone, walking upstairs or downstairs in low-light conditions while using the phone, walking on slippery surfaces while using the phone, walking in low-light environments while using the phone, walking in pedestrian crossing areas while using the phone, standing still on stairs while using the phone, standing still on slippery surfaces while using the phone, standing still in low-light environments while using the phone, and standing still in pedestrian crossing areas while using the phone.

Then, we recruited 30 volunteers to collect experimental data for the nine aforementioned categories using mobile phones of various brands. The specific data collection parameters were as follows: the sampling frequency was set at 50 Hz to capture time series data of X-, Y-, and Z-axis accelerations. Simultaneously, the rear camera of the mobile phone was utilized with a filming rate of 30 frames per second (fps) to capture video footage of the volunteer’s surrounding environment at the time.

The process of producing “image-acceleration” multimodal data pairs is as follows: Firstly, we crop and align the collected three-axis acceleration time series and video data based on time. Then, using a fixed length sliding window, the acceleration and video datasets are segmented and matched into standard multimodal data samples that meet the requirements of this study. We set the sliding window length to 6 s and the moving step size to 3 s. We ensure that there is a 50% overlap between adjacent data segments, which is beneficial for preserving the continuity and trend of changes between the data. Finally, we extract the endpoint time keyframes from the segmented video data to complete multimodal data matching between the acceleration time series and a single-frame image. The matched multimodal sample data consist of a 6 s acceleration time series and an image. The alignment rules for the timeline are illustrated in Figure 3.

In this work, the experimental dataset comes from 9000 pairs of data collected by 30 volunteers, with each volunteer collecting 300 pairs of data. Each pair of multimodal data includes an image and corresponding 6 s acceleration time series data. Then, based on 9000 pairs of experimental data, we designed the following cross-validation experiments: 900 pairs of multimodal data are randomly selected from 3 people as a group, and 9000 pairs of multimodal data from 30 people are divided into 10 groups. The training set, validation set, and test set are divided in a 7:2:1 ratio, and each group is used as the test set in sequence. The remaining nine groups are randomly divided into seven training sets and two validation sets. The experimental results were evaluated using the average of 10 sets of test sets as the final evaluation result. Some of the experimental samples are shown in Figure 4. Furthermore, we also considered the balance of sample numbers among various categories to prevent the model from overfitting to specific categories of data.

## 4. Multimodal Phubbing Danger State Recognition Network

The architecture of the multimodal dangerous state recognition network for image-acceleration in this text is illustrated in Figure 5. The model mainly consists of three parts: the user’s surrounding environment feature extraction module, the user’s motion state feature extraction module, and the feature fusion and danger recognition module. Convolutional neural networks (CNNs) have gained widespread applications in areas such as image classification, object detection, and so on. Their network architectures have also been continuously improved, leading to the development of various classic models, such as VGG [16], GoogleNet [17], ResNet [18], DenseNet [19], and others. Although the accuracy of the models has been effectively improved, the computational complexity and model size have also significantly increased accordingly. However, in the embedded environment of mobile phones, the limited processor performance, memory capacity, and power consumption restrict the use of high-precision network models to meet real-time requirements on mobile phones. To address the above issue, this paper adopts the depth-wise separable convolution module from the ultra-lightweight network MobileNet [20] to construct the main backbone networks for feature extraction in each branch.

### 4.1. Surrounding Environment Feature Extraction Module

In the surrounding environment feature extraction module, we achieve feature extraction of the input scene RGB images by concatenating depth-wise separable convolution modules. The backbone network consists of five blocks, as shown in the pink area in Figure 5. In Block1, there is a traditional convolution with a 3 × 3 kernel and a stride of 2, as well as a depth-wise separable convolution. In the separable convolution, both the 3 × 3 depth-wise convolution and the 1 × 1 point-wise convolution have a stride of 1. Block2 only contains a depth-wise separable convolution block with a stride of 2. Block3 and Block4 have the same structure, each consisting of two depth-wise separable convolutions concatenated with strides of 1 and 2, respectively. Block5 is composed of seven depth-wise separable convolution blocks concatenated together. To minimize computational load and reduce model training time, we utilize a resolution factor (β=6/7) to downsample the original 224 × 224 RGB environmental image to 192 × 192, achieving accelerated computation. The output of Block5 is a 1024-channel, 6 × 6 environmental feature map. Through an average pooling layer, the feature map is dimensionally reduced, and then flattened to obtain a one-dimensional environmental feature descriptor vector with a length of 1024.

### 4.2. Motion Feature Extraction Module

#### 4.2.1. GAF Image Encoding and Feature Extraction

To effectively fuse with the environment features based on image data, we map the one-dimensional time series of gravity acceleration, representing the user’s behavior, into the two-dimensional feature space of a Gram Angular Field image. The Gramian Angular Field (GAF) image encoding method was proposed by Wang et al. [21] in 2015. Assuming a given time series $X=\{x_{1},x_{2},\ldots ,x_{n}\}$, we utilize a normalization method to process *X*, ensuring all values fall within the interval [−1, 1]. This prevents the inner product from being biased towards the observation with the maximum value. The normalization formula is as follows:(1)$${\tilde{x}}_{-1}^{i}=\frac{{(x}_{i}-max(X))+(x_{i}-min(X))}{\max\left(X\right)-\min\left(X\right)}$$

After normalizing the data in the above equation, the phase angle in polar coordinates is obtained by taking the arccosine function. The normalized time series data, represented as $\tilde{X}$, are encoded in polar coordinates with the corresponding timestamps as radii. The calculation formula is as follows:(2)$$\left\{\begin{array}{c} \emptyset =\arccos\left({\tilde{x}}_{i}\right),{-1\leq \tilde{x}}_{i}\leq 1,{\tilde{x}}_{i}\in \tilde{X}\\ r=\frac{t_{i}}{N},t_{i}\in N \end{array}\right.$$

In the equation, $t_{i}$ represents the time step, and *N* is a constant factor. The encoding mapping function based on Equation (3) has an important property: it is a bijective mapping function. Because when ∅∈[0,π], the function cos⁡∅ monotonically decreases, the proposed mapping function generates one and only one result with unique inverse mapping in the polar coordinate system.

After transforming the time series into polar coordinates, the sine function value of the phase angle differences between each feature point can be used to describe the correlation between feature points at different time intervals. The definition of the Gramian Angular Difference Field (GADF) is as follows. In the equation, *I* is the unit row vector.
(3)$$GADF=\left[\begin{array}{ccc} \sin\left(\emptyset_{1}-\emptyset_{1}\right) & \cdots & \sin\left(\emptyset_{1}-\emptyset_{n}\right)\\ \sin\left(\emptyset_{2}-\emptyset_{1}\right) & \ldots & \sin\left(\emptyset_{2}-\emptyset_{n}\right)\\ \vdots & \ddots & \vdots \\ \sin\left(\emptyset_{n}-\emptyset_{1}\right) & \cdots & \sin\left(\emptyset_{n}-\emptyset_{n}\right) \end{array}\right]={\sqrt{I-{\tilde{X}}^{2}}}^{\prime }\cdot \tilde{X}-{\tilde{X}}^{\prime }\cdot \sqrt{I-{\tilde{X}}^{2}}$$

According to the definition of the GADF matrix given in Equation (3), the elements of this matrix are the sine function values of the differences between any two phase angles established by Equation (3) in the polar coordinate system. And from the first row to the nth row, and from the first column to the nth column, as the phase angle difference ($∆\emptyset =\emptyset_{i}-\emptyset_{j}$) increases, the time interval of data *X* also increases. Thus, the temporal dimension of variable $x_{i}$ is encoded into the geometric structure of the matrix defined in Equation (3).

The following two factors are considered: (1) The mutual relationship and variation patterns between data from each coordinate axis contribute to the description of complex motion states of mobile phone users. (2) For microprocessors targeted towards mobile phones, high-resolution and multi-channel image representations will affect computational complexity and processing speed. In this paper, the three-axis acceleration time series are compressed into one-dimensional time series vectors. Five forms of dimensionality reduction are employed, including subtraction of each axis datum, subtraction of three axes (x-y, x-z, y-z, x-y-z), and the arithmetic square root, the formula for which is as follows:(4)$$V=\sqrt{{a_{x_{i}}}^{2}+{a_{y_{i}}}^{2}+{a_{z_{i}}}^{2}}$$

In the equation, $a_{x_{i}},$ $a_{y_{i}}$ and $a_{z_{i}}$ represent the accelerations corresponding to the X, Y, and Z axes, respectively.

In Figure 6, the left side shows the original three-axis acceleration signals of different behaviors and five reduced dimensional acceleration time series signals in the training set. Combining the mean evaluation results of the signal energy based on five different dimensionality reduction methods for different behaviors in the training set shown in Table 1, it can be observed that X-Y-Z signals have the highest energy. This indicates that the degree of oscillatory variation in the dimensionality reduction signal over time is most significant. In other words, the representation capability for different behaviors of users is the strongest. Therefore, this work selects X-Y-Z signals as inputs to the GADF encoder, and the examples of GADF features are provided on the right.

Next, we input the transformed X-Y-Z dimensionality-reduced signals’ GADF pseudo-images into the lightweight feature extraction network MobileNet. Due to the relatively simple structure of GADF pseudo-images, compared to RGB images in natural scenes, their grayscale histogram distribution and spatial distribution of texture features are simpler, more intuitive, and more regular. Therefore, to prevent over-extraction of features and reduce computational complexity, we use a width factor (α = 0.75) to decrease the number of feature channels extracted in the neural network. Finally, after average pooling and flattening, we obtain a one-dimensional feature vector of length 768.

#### 4.2.2. Statistical Feature Extraction

In order to enhance the representation capability of features without increasing the computational complexity of feature extraction, we extract statistical features from the X-Y-Z acceleration time series using nine statistical models defined by Equations (5)–(13), including mean, variance, maximum value, minimum value, median, skewness, kurtosis, interquartile range, and coefficient of variation. The calculation formulas are as follows:(5)$$\mu =\frac{1}{n}\sum_{i=1}^{n}x_{i}$$
(6)$$\sigma^{2}=\frac{1}{n}\sum_{i=1}^{n}{\left(x_{i}-\mu \right)}^{2}$$
(7)$$MAX\left(X\right)=\max\left(x_{1},x_{2},\ldots ,x_{n}\right)$$
(8)$$MIN\left(X\right)=\min\left(x_{1},x_{2},\ldots ,x_{n}\right)$$
(9)$$Median\left(X\right)=\left\{\begin{array}{cc} x_{\frac{n+1}{2}} & n\;is\;odd\\ \frac{x_{\left(\frac{n}{2}\right)}+x_{\left(\frac{n}{2}+1\right)}}{2} & n\;is\;even \end{array}\right.$$
(10)$$Skewness\left(X\right)=\frac{n}{\left(n-1\right)\left(n-2\right)}\sum_{i=1}^{n}{\left(\frac{x_{i}-\mu }{\sigma }\right)}^{3}$$
(11)$$Kurtosis\left(X\right)=\frac{n(n+1)}{\left(n-1)(n-2)(n-3\right)}{\sum_{i=1}^{n}(\frac{x_{i}-\mu }{\sigma })}^{4}-\frac{3{\left(n-1\right)}^{2}}{\left(n-2\right)\left(n-3\right)}$$
(12)$$IQR=Q_{3}-Q_{1}$$
(13)$$C_{v}=\frac{\sigma }{\mu }$$
where n represents the total number of data points in each segment, $x_{i}$ represents the gravitational acceleration value collected at the i-th point, σ represents the standard deviation, and $Q_{1}$ and $Q_{3}$ respectively represent the data points located at the 25th and 75th percentiles after sorting the data in ascending order. To balance the dimensional difference between deep learning features and statistical features, the aforementioned 63 statistical metrics are composed into a one-dimensional feature vector. Subsequently, a fully connected layer is utilized to expand it into a one-dimensional statistical vector of length 256.

#### 4.2.3. Fusion of Deep Features and Statistical Features

The 1 × 256 statistical features are concatenated with the 1 × 768 deep features to form a fused feature vector of length 1 × 1024. To optimize the effectiveness of each component in the fused feature vector, the Gated Linear Unit (GLU) [22] is utilized to modulate the concatenated fused feature information. Its calculation formula is
(14)$$GLU\left(X\right)=X⊗\sigma \left(XW_{g}+b_{g}\right)$$

In the equation, X is the input vector, σ is the sigmoid activation function, ⊗ denotes the Hadamard product, and $W_{g}$ and $b_{g}$ are the weights and biases of the gating layer.

### 4.3. Multimodal Feature Fusion for Surrounding Environment and Behavior

First, the environment feature vector of length 1024 is concatenated with the motion state feature vector to obtain a combined feature vector of length 1 × 2048. Next, the SENet [23] channel attention model is incorporated to evaluate the contribution of each feature component to the representation capability by calculating importance weights. And these weight values are utilized to adaptively adjust feature components, enabling the network model to focus more on features crucial for classification tasks, thereby enhancing the model’s expressive power and generalization ability. Finally, the extracted features are mapped to nine phubbing danger state categories through fully connected layers and the SoftMax function.

### 4.4. Loss Function and Optimizer

In the training and validation loop of the model, the cross-entropy loss function is used to calculate the difference between the model’s predictions and the true labels. In this task, the cross-entropy loss function penalizes samples whose predicted results differ significantly from the true results. The calculation formula is
(15)$$L=-\frac{1}{N}\sum_{i=1}^{N}\sum_{c=1}^{M}y_{i,c}\log\left(p_{i,c}\right)$$

In the equation, N is the number of samples, M is the number of classes, $y_{i,c}$ represents the true label of sample i in class c (0 or 1), and $p_{i,c}$ represents the predicted probability that the model assigns sample i to class c.

Meanwhile, the SGD optimizer with momentum is used to update the model parameters to minimize the loss function. The momentum term can help accelerate convergence and prevent getting stuck in local optimality. Its update formula is
(16)$$\upsilon_{t}=\gamma \upsilon_{t-1}+\eta \nabla_{\theta }L\left(\theta \right)$$
(17)$$\theta =\theta -\upsilon_{t}$$
where $\upsilon_{t}$ is the momentum, γ is the momentum coefficient, η is the learning rate, and $\nabla_{\theta }L\left(\theta \right)$ is the gradient of the current parameter θ.

Combining the cross-entropy loss function and the SGD optimizer, the model continuously optimizes its parameters by calculating the loss and updating the parameters in each iteration, thereby improving the model’s performance during training.

## 5. Experimental Results and Analysis

### 5.1. Experimental Environment and Evaluation Metrics

The experimental environment for model training and testing in this paper includes both offline and online components. The quantitative performance testing of the model is conducted entirely in the offline environment, while we only demonstrate the applicability and stability of the model on the phone in the online environment. The specific experimental environment configurations for the offline training server and online testing mobile devices are shown in Table 2.

The offline model training was conducted on a Windows 10 (Education Edition) server, using the PyTorch framework to train and generate a “.pt” model. The “.pt” model was then converted into a “.tflite” model for use by TensorFlow Lite. TensorFlow Lite is a lightweight solution, typically optimized for performance on mobile devices, designed specifically for low-latency and low-power scenarios. Finally, the “.tflite” model was deployed to a mobile phone via Android Studio, enabling real-time online detection of dangerous states for “phubbers”.

We adopt two metrics, accuracy and *F*1-score, to evaluate the model’s accuracy. The specific calculation formulas are as follows:(18)$$Accuracy=\frac{TP+TN}{TP+TN+FP+FN}$$
(19)$$F_{1}=\frac{2\times precision\times recall}{precision+recall}$$
*TP* and *TN* represent the number of samples correctly predicted as positive and negative classes, respectively. *FP* and *FN* represent the number of samples incorrectly predicted as positive and negative classes, respectively.

### 5.2. Experimental Validation of Input Signal Effectiveness for GADF Pseudo-Image Generator

In order to effectively describe mobile user behavior using GADF features while minimizing computational complexity, we compared different dimensionality reduction methods for combinations of raw accelerometer data from the X, Y, and Z axes. These methods include V (Equation (5)), X-Y, X-Z, Y-Z, and X-Y-Z. And the identification results of different dimensionality reduction methods are summarized in Table 3.

By observing Table 3, it can be seen that after dimensionality reduction using the V, X-Z, and X-Y-Z methods, the corresponding model recognition accuracy indicators, accuracy and F1-score, are higher, indicating better performance compared to directly utilizing pseudo-image features generated from raw X-, Y-, and Z-axis time series data separately. Among these, the performance of X-Z and X-Y-Z is significantly better than that of other methods. In particular, the dimensionality reduction method X-Y-Z corresponds to the highest model recognition performance, with an accuracy of 0.9598 and an $F_{1}$-score of 0.9775. The experiment validated the effectiveness of using the energy indicators of the signal to evaluate and select the dimensionality reduction method as described in Section 4.2.1. Figure 7a,b respectively present confusion matrices of the test results for the model with GADF input as X-Z and X-Y-Z.

By comparing Figure 7a,b, it can be seen that the two models each have their strengths and weaknesses in recognizing different categories. Specifically, the X-Z model exhibits a lower testing accuracy of 90.0% in the category of CDAR (climbing stairs + darkness), with 10.0% falsely detected as category WDAR (walking + darkness). Meanwhile, the X-Y-Z model demonstrates the lowest testing accuracy of 77.8% in the category of SSTA (static + stairs), with 14.4% and 7.8% mistakenly classified as category SDAR (static + darkness) and category SZEB (static + zebra crossing), respectively. Therefore, the following conclusion can be drawn: The X-Z model cannot effectively distinguish between walking and going up and down stairs in low-illumination environments. The X-Y-Z model is more likely to confuse stairway areas under static conditions with low-lighting environments and zebra crossing areas. Overall, the average accuracy across the nine categories is higher for the X-Y-Z model compared to the X-Z model.

### 5.3. Ablation Experiment

To validate the effectiveness of multimodal observation data combining images and accelerations for identifying the hazardous states for “phubbers”, evaluate the representational capabilities of each modality, and analyze potential complementary or redundant relationships between different modalities, we decompose and recombine the multimodal hazardous state recognition network illustrated in Figure 5 into different architectures, and evaluate the performance changes in each structure individually. The experimental results are shown in Table 4.

From Table 4, the following can be observed: (1) The accuracy of the three recognition models based on multimodal observation data combining images and accelerations is superior to the four recognition models based on single-modal observation data of either images or accelerations. This indicates that neither the environmental information collected by the rear-view camera nor the user behavior information collected by the gravity acceleration sensor can provide accurate and sufficient representation capabilities for identifying hazardous states for “phubbers”. Conversely, after the effective integration of image and acceleration modal information, the characteristics of the environment and user behavior can complement each other, describing the dangerous features of the behavior of “phubbers” from multiple perspectives, thereby achieving higher recognition accuracy. (2) Additionally, we conducted an effectiveness analysis of the fusion between RGB image features with accelerometer time series features obtained from different feature encoding methods. Through comparison, it is evident that, as shown in Figure 5, it enables the utilization of statistical features to complement the information loss in GADF pseudo-images, thereby enhancing the robustness and accuracy of the recognition model.

### 5.4. Comparative Experiment

To verify the effectiveness of the proposed mobile-oriented multimodal phubbing risk recognition network in this paper, the performance of this model is compared and analyzed with existing phubbing risk recognition models on the same dataset across four dimensions: accuracy, parameter volume, execution time, and model storage size. The comparison includes three single-modal models (SMCNN [24], Extra-Trees [25], and Dempster–Shafer [7]) and two multimodal models (ResNet [26] and 2D CNN-LSTM [27]), with the experimental results shown in Table 5.

By comparison, it is evident that multimodal recognition methods generally outperform single-modal recognition methods in terms of accuracy. This advantage can be attributed to the ability of multimodal models to effectively fuse multi-dimensional features from different modal data, thereby enhancing recognition accuracy through modal complementarity. Additionally, the fusion of multimodal features implies higher model complexity and computational overhead. Among the three multimodal recognition models presented in Table 5, the ResNet network based on RGB-GADF image data achieves the highest recognition accuracy, reaching 0.9635. However, the complex network structure results in the largest parameter size, largest model storage size, and slowest computational speed. On the contrary, although the accuracy of our model is 0.9598, which is decreased by 0.37% compared to ResNet, it exhibits the smallest parameter volume, fastest computational speed, and lowest model storage size among all models listed in Table 5. This presents a significant advantage for deployment in the constrained computational and memory resources of the Android mobile smartphone environment. In practical applications, we not only pursue accuracy in detecting hazardous states for “phubbers” but also ensure the timeliness of warning alerts, achieving effective warning protection for them.

### 5.5. System Performance Testing Based on Mobile Smartphones

Based on the multimodal image-acceleration network proposed in this paper, we developed a smartphone application for intelligent recognition and warning of hazardous states for phubbers. The application interface is shown in Figure 8. The application interface is divided into two parts: the upper part displays the real-time road conditions captured by the rear-view camera, while the lower part shows the curves of the X-, Y-, and Z-axis acceleration data as they change over time during user movement. When “phubbers” are detected to be in a hazardous state, we display a danger alert in red font at the top and simultaneously issue a voice warning, creating a composite warning design combining visual and auditory features. When the user is in a safe state, the word “Safe” is displayed in green, and no voice prompt is issued.

In order to alleviate the rapid power consumption and potential user privacy issues caused by long-term use of cameras, we used the 63 statistical features extracted in Section 4.2.2 to train a decision tree model as the user behavior analysis module, as shown in the green box in Figure 1. When the user’s activity at the moment is recognized as using the phone in a stationary state, walking on flat ground while using the phone, or walking upstairs or downstairs while using the phone, the rear camera of the phone is activated to capture images of the user’s surroundings, and the proposed multimodal network is used for hazard identification to achieve adaptive energy-saving control of mobile phone cameras. After offline testing, the accuracy of the decision tree model is 92.36%, and the computational speed reaches 0.02 s.

Figure 9 presents the experimental results of the mobile application for the proposed multimodal phubbing danger state recognition network based on images and acceleration. The test scenarios encompassed various real-world instances, as shown below. Figure 9a depicts a user walking on damp and slippery surfaces while engrossed in their phone. Figure 9b illustrates a pedestrian crossing a busy intersection with their attention diverted to the screen. Figure 9c,d showcase a user ascending and descending stairs, respectively, while distracted by their phone. Figure 9e depicts a user engaging with their phone in a dimly lit environment. Figure 9f portrays a person sitting down, engrossed in their phone.

## 6. Conclusions

Accidents caused by playing with mobile phones while looking down in outdoor environments include falling into the water, crashing into a car, and losing children. These accidents not only pose a threat to personal safety, but may also cause unpredictable harm to those around the user. The mobile-based phone addicts’ danger detection and warning method has become the mainstream method for preventing such accidents due to its advantages of not requiring other auxiliary devices and providing real-time monitoring. However, existing methods all use camera images from mobile phones as a single observation datum. Assuming that the current user is playing with the phone, potential danger warnings are made by recognizing the scene’s environment information in the images, such as intersections and staircase areas. This method has obvious shortcomings, as it ignores the determination of the timing of camera activation. Therefore, it can lead to missed warning opportunities for danger, or rapid battery depletion of mobile phones caused by constantly turning on the camera.

In response to the above issues, this article proposes for the first time a multimodal phone addiction hazard identification network model and early warning system. Different from existing methods, we combine the behavior analysis of mobile phone users with the recognition of the surrounding environment. Real-time monitoring of user behavior while playing with mobile phones using acceleration data and driving the activation of the phone camera through behavior recognition results in achieving energy-saving control of the camera and reducing battery loss for the phone. Meanwhile, we propose a lightweight “acceleration-image” multimodal hazard identification network model, which significantly enhances the recognition results for hazardous states while ensuring real-time performance. The specific implementation method is as follows: To reduce the complexity of the neural network model and ensure real-time performance, a lightweight MobileNet is adopted as the backbone network for image feature extraction. To address the heterogeneity between 2D images and 1D accelerometer time series data, we utilize a GADF pseudo-image encoder to map the one-dimensional temporal data into the two-dimensional image space. Simultaneously, we utilize statistical features to enhance the representational capacity for behavioral features.

The experimental results show that based on the self-made dataset, the proposed model has an average running time of 0.08 s and an average recognition rate of 95.98%. Compared to existing single-modality hazard identification models based on environmental recognition, the accuracy is improved by more than 6%; compared to the dual-branch ResNet and CNN-LSTM multimodal network models, the computation time is reduced to less than 25%. Compared to existing methods, the multimodal hazard identification network proposed in this article achieves a good balance between computation time and accuracy, making it more suitable for deployment on mobile devices to achieve real-time hazard detection and warning functions for phone addicts.

In future research, we will attempt to address the practicality issues with mobile applications, including the application running in the background, pop-up warnings, camera occupancy detection, and other functions, to ensure that users can use their phones normally while achieving danger state recognition and receiving warnings. In addition, we will focus on researching and addressing personal privacy issues that may arise when using cameras, such as by increasing the blocking of private places and portraits, indoor and outdoor environment recognition, and cleaning historical image data.

## Figures and Tables

**Figure 1 sensors-24-04654-f001:**
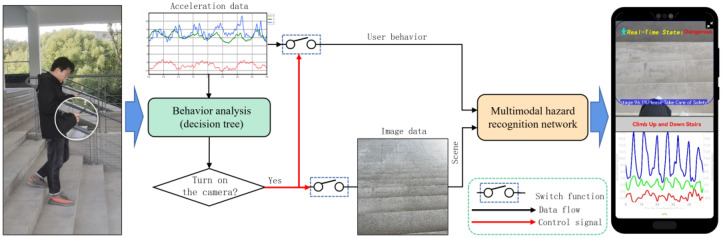
Framework of multimodal phubbing danger detection system.

**Figure 2 sensors-24-04654-f002:**
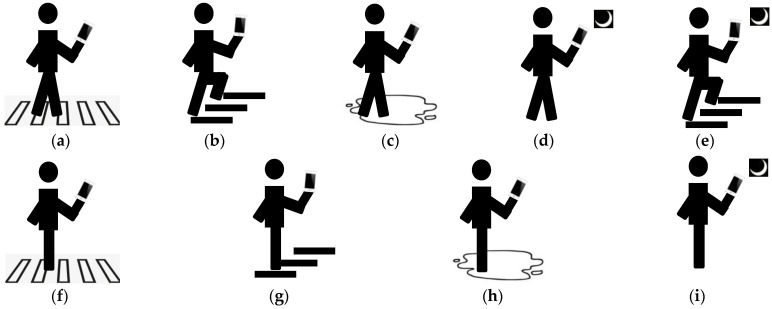
Examples of the nine dangerous states defined in this paper. (**a**) walking and zebra crossing. (**b**) climbing stairs. (**c**) walking and wet surface. (**d**) walking and darkness. (**e**) climbing stairs and darkness. (**f**) static and zebra crossing. (**g**) static and stairs. (**h**) static and wet surface. (**i**) static and darkness.

**Figure 3 sensors-24-04654-f003:**
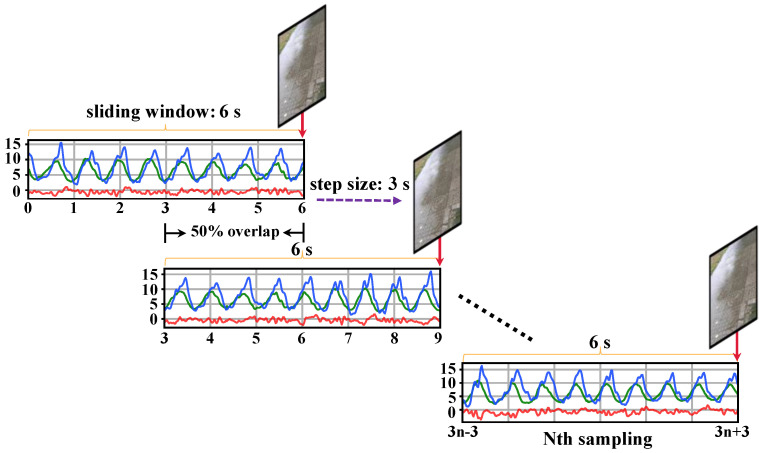
Rules for pairing environment images with sensor data. In the sensor data, the red, green, and blue lines represent the X-axis, Y-axis, and Z-axis, respectively.

**Figure 4 sensors-24-04654-f004:**
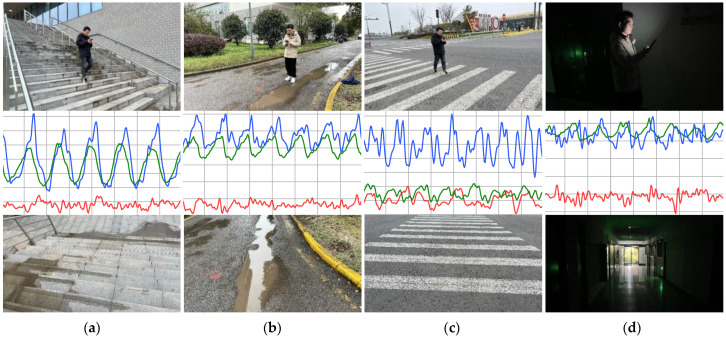
Data collection sample images. The first row shows the actual scene where the user is located, while the second and third rows depict the corresponding sensor time series and environmental real-life images. The sensor time series for the X, Y, and Z axes are represented by red, green, and blue lines, respectively. (**a**) climbing stairs. (**b**) walking and wet surface. (**c**) walking and zebra crossing. (**d**) walking and darkness.

**Figure 5 sensors-24-04654-f005:**
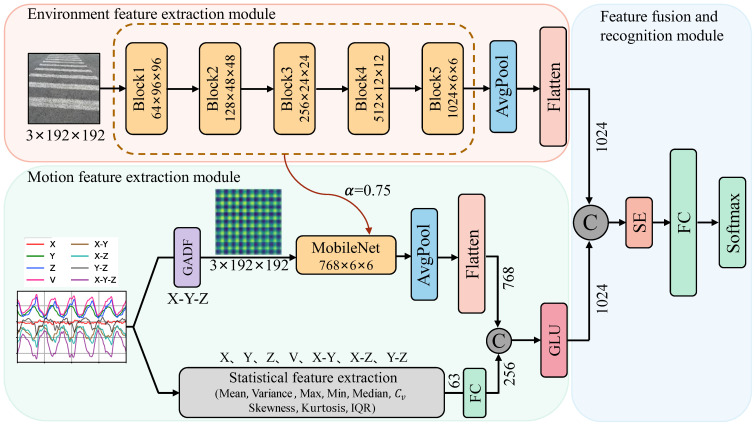
Multimodal phubbing danger state recognition network.

**Figure 6 sensors-24-04654-f006:**
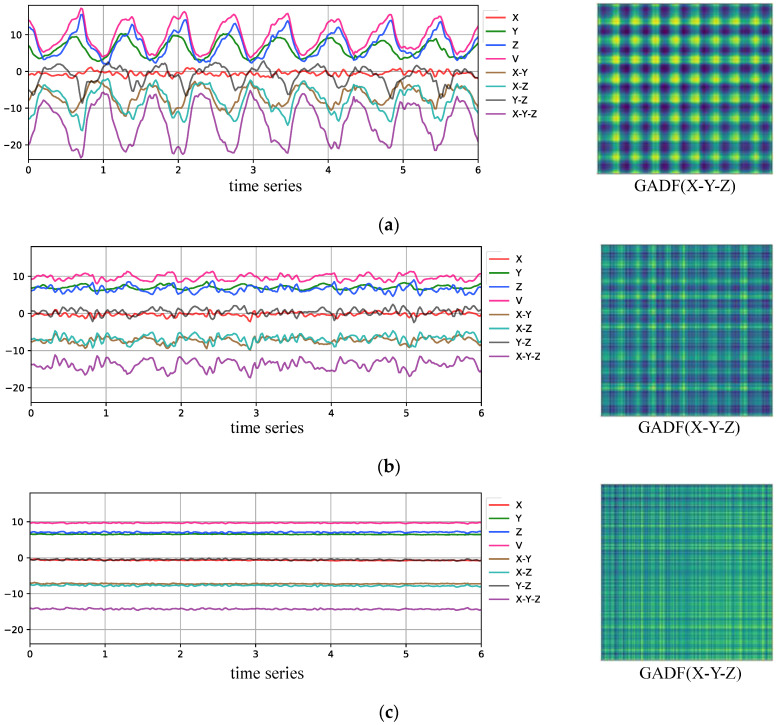
Sensor curves and GADF pseudo-images corresponding to different behaviors. (**a**) Sample of sensor curve and X-Y-Z pseudo-image when going up and down stairs. (**b**) Sample of sensor curve and X-Y-Z pseudo-image when walking. (**c**) Sample of sensor curve and X-Y-Z pseudo-image when stationary.

**Figure 7 sensors-24-04654-f007:**
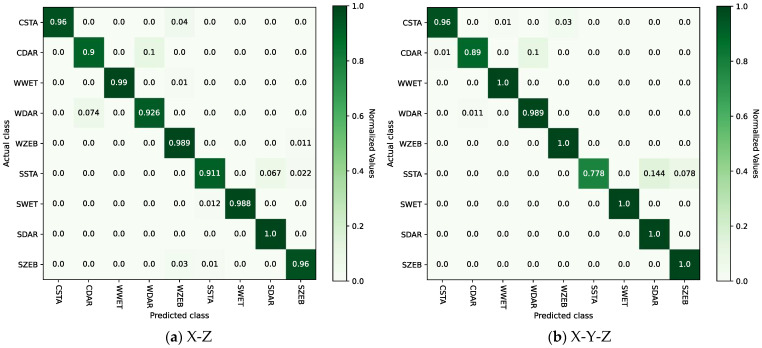
Confusion matrices of test results with GADF input as X-Z and X-Y-Z.

**Figure 8 sensors-24-04654-f008:**
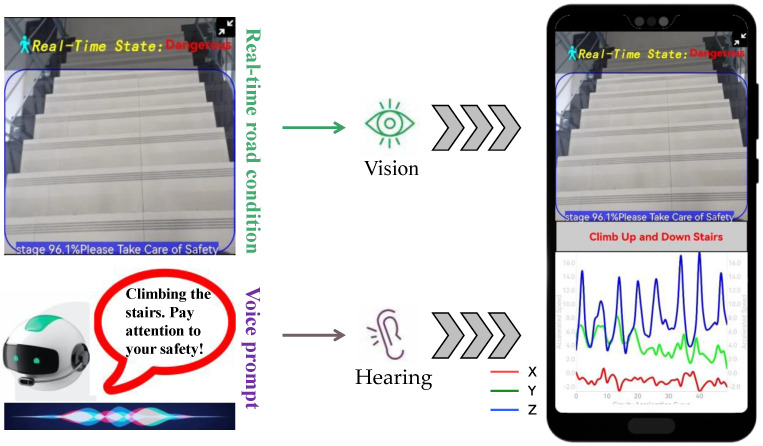
Mobile application user interface.

**Figure 9 sensors-24-04654-f009:**
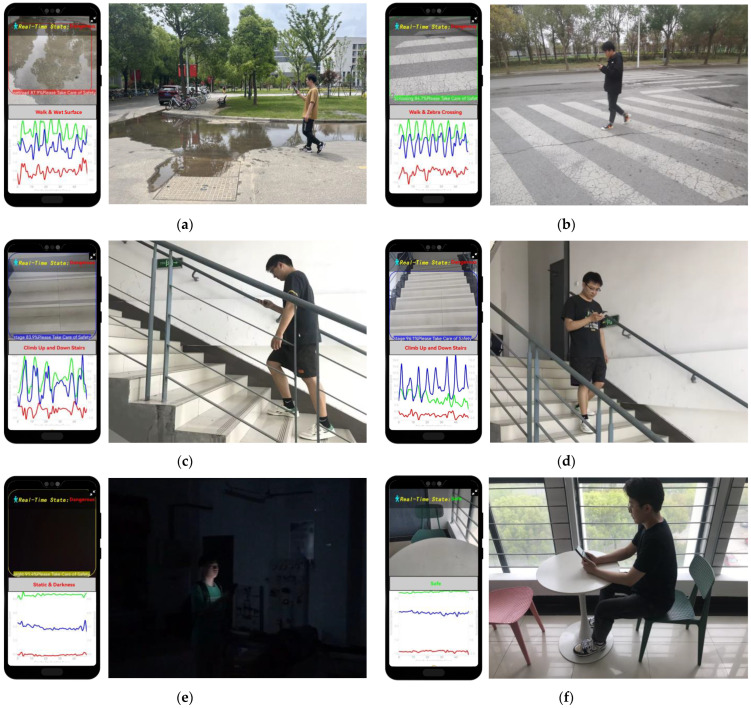
Examples of online test results for mobile phone app. The left side of each figure shows the real-time display of the mobile app, while the right side shows the actual photos of the user’s activity state and environment. (**a**) Walking and wet surfaces. (**b**) Walking and zebra crossing. (**c**) Going upstairs. (**d**) Going downstairs. (**e**) Static and darkness. (**f**) Sitting and browsing on the phone.

**Table 1 sensors-24-04654-t001:** Signal energy evaluation of different dimensionality reduction methods.

	V	X-Y	X-Z	Y-Z	X-Y-Z
Climbing up and down stairs	112.63	52.28	79.38	8.46	232.29
Walking	96.23	52.53	47.40	1.13	193.20
Static	95.02	52.02	66.36	0.93	209.66

**Table 2 sensors-24-04654-t002:** Experimental environment.

Server Parameters	Specifications	Mobile Device Parameters	Specifications
CPU	I5-12490F	Type	HUAWEI Mate30 (HUAWEI, Shenzhen, China)
GPU	Nvidia-RTX3060	Processor	Kirin 990 5 G
RAM	16 GB	RAM	8 GB
Operating system	Windows 10	Operating system	Android 12
IDE	Pycharm 2022.3.2	--	--
Code language	Python 3.8.0	--	--
Network framework	Pytorch 1.11.0	--	--
CUDA	11.3	--	--
cuDNN	8.2.0	--	--

**Table 3 sensors-24-04654-t003:** Test results under different inputs to GADF.

Input to GADF	Accuracy	$F_{1}$-Score
X	0.8750	0.9132
Y	0.9301	0.9586
Z	0.9396	0.9603
V	0.9479	0.9708
X-Y	0.8823	0.9184
X-Z	0.9574	0.9758
Y-Z	0.9135	0.9411
X-Y-Z	**0.9598**	**0.9775**

**Table 4 sensors-24-04654-t004:** Ablation study results.

Input	Accuracy	*F*_1_-Score
RGB	0.5398	0.5585
GADF	0.5917	0.6149
Statistical features	0.5066	0.5265
GADF + statistical features	0.6151	0.6253
RGB + GADF	0.9210	0.9521
RGB + statistical features	0.8865	0.9016
Ours	**0.9598**	**0.9775**

**Table 5 sensors-24-04654-t005:** The comparison results for accuracy and parameters for different networks.

Type	Input	Model	Accuracy	*F*_1_-Score	Parameters	Time (s)	Size (Mb)
Single-modal	ATS	SMCNN [24]	0.8983	0.9117	11,190,537	0.12	42.76
	ATS	Extra-Trees [25]	0.7765	0.7894	2,767,110	0.05	25.41
	ATS	Dempster–Shafer [7]	0.8759	0.8893	3,525,126	0.06	27.10
Multimodal	RGB + GADF	ResNet [26]	**0.9635**	**0.9792**	22,362,249	0.47	85.46
	RGB + ATS	2D CNN-LSTM [27]	0.9536	0.9678	12,376,284	0.31	56.96
	RGB + GADF + statistic	Ours	0.9598	0.9775	6,874,567	**0.08**	**28.36**

## Data Availability

The datasets used and/or analyzed during the current study are available from the corresponding author upon reasonable request.

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
