# Peer review of "Image-Acceleration Multimodal Danger Detection Model on Mobile Phone for Phone Addicts"

_sensors, 2024, doi:10.3390/s24144654_

Round 1

Reviewer 1 Report

Comments and Suggestions for Authors

This paper reports the design and experimental evaluation of a multi-modal danger detection model based on video frames and accelerometer readings. The results show that the method is comparable to other state-of-the-art models, relying on a smaller and faster model. An ablation study is included, showing complementarity among features from different modalities. It was evaluated on a relatively large dataset. The model was also successfully deployed as a mobile application.

General Comments

The text states that 8550 sets of experimental samples were collected from 29 volunteers. This means that it is highly likely that more than one class-person combination was collected. It also states that the training, validation, and test sets were sampled in a ratio of 7:2:1. However, the text does not make it clear how the data sampling was performed. A person's gait is rather unique. Thus, the sensor readings from a person are very specific. Thus, if data from the same person is in both training and test sets, the model may correctly predict the output not because it generalized well but because it learned the patterns for specific people very well. Thus, my primary concern is that the experiments are not evaluating model generalization properly. Considering this, I would advise resampling the training, validation, and test sets, making sure that experimental samples from the same person are not in both training and test sets. The StratifiedGroupKFold in sklearn does this in a very simple manner.

How was the process of constructing the model? Did you use any architectural tuning tools? Please report it in the text.

This seems like a very interesting dataset. It would be nice if you could make it available to the research community. It would simplify the comparison with the works of others. It also increases the credibility of the results since other researchers can replicate and validate your work.

Specific Comments

l. 83 "However, single-modal models for identifying "phubbing" behavior have lower accuracy rates." - how do you know this? isn't this what you are trying to show with your experiments?

l. 131-139 and Fig 3 - From what I could understand from the Figure, the frame corresponding to the last accelerometer sample is used as a reference, and then the remainder of the frames are selected backward, 3s apart. This is not clear from the text, especially regarding the 3s interval. Please make this part clearer, as it is central to your contribution.

l. 140-147 - Also related to the previous comment, each "experimental sample" corresponds to how much time? Is it 15s (based on Figure 3)? Please make this clearer in the text.

l. 238-246 and Table 1 - What data did you use to calculate the signal energy to select the most appropriate combination (X-Y-Z)? Only the training data should be used for this, as it is considered a hyperparameter of the model. Update the text explaining what data was used. If the entire dataset was used, you should re-run the experiments.

The results presented in sections 4.2, 4.3, and 4.4 were obtained with the offline, test dataset or using the app? I believe it was using the offline test dataset (described in lines 142-144), but the text does not specify this. My confusion comes from the fact that the first paragraph of section 4.1 states, "The experimental environment for training and testing the models in this paper includes two parts: offline training environment and online testing environment. The offline model training is conducted on a Windows 10 server, resulting in the generation of PT models. Subsequently, the PT model is converted into a TF model and deployed to mobile phones through Android Studio, enabling real-time online detection". From what I understand, this text implies that all model testing was done online, but I can't be certain from the text. Please make the text clearer in this regard, explicitly stating what was used as the test data to generate the results.

Figure 7: The x-axis should be "Predicted class". It would also be nice to give a code to each class so the reader can more readily recognize what class each line/column represents. For instance, WZEB (walking + zebra), SZEB (static + zebra), and so on.

l. 378, 391, 394 - "model capacity" is ambiguous. Use "model storage size" instead.

l. 442 -  "the model proposed in this paper exhibits higher accuracy" - higher accuracy than what? Please make it clearer.

Comments on the Quality of English Language

The quality of the English language in this paper is appropriate, but some minor flaws (reported below) detract from the reading experience.

l. 142, 244 "This Paper" - the paper is the report of your work. "This does not "collect" or "select" anything. Please use a more suitable term such as "in this work".

l. 118 to 127 - "playing with phone" and "playing the phone" sentences are awkward in this context. Replace with something such as "using the phone".

Fig 2. I don't recognize the term "hydrops". Maybe "water accumulation" or "wet floor" are better alternatives.

Fig. 5 "MoblieNet" is incorrect. 

l. 301,302 - "PT model" - what does PT stand for? Is it pytorch? Please define this in the text.

l. 302 - "TF model" - what does TF stand for? Is it tensorflow? Please define this in the text.

l. 305 - "in the following Table 2" - it should be "in Table 2"

l. 407, 408 "creating a stereoscopic warning design combining visual and auditory" - the word stereoscopic is used inappropriately here. This sentence is also confusing, although I understand its meaning. Please rewrite for clarity.

l. 440  "statistical feature" should be "statistical features"

The word "terminal" is used as a synonym for a phone throughout the text. I think it would be better to use the term "phone" instead since "terminal" never means anything other than phone.

Reviewer 2 Report

Comments and Suggestions for Authors

 The authors propose a multimodal danger perception network model that combines the smartphone's rear camera and built-in gravity acceleration sensor to warn of potential dangers for phubbers. The authors validated the model's performance on a self-built database.

1. I am concerned about the practicality of this algorithm, as few people are willing to grant an app real-time access to their camera around the clock. It is very unfriendly to privacy protection. How do the authors plan to address this issue?

2. Literature review should be written in a separate section.

3. The authors validated the performance using a self-built database in the experimental phase. It is recommended to validate the model's performance on additional databases to enhance its persuasiveness.

Comments on the Quality of English Language

Moderate editing of English language required

Reviewer 3 Report

Comments and Suggestions for Authors

 a)      Due to subjectivity, I do not recommend using the word “Phubbers” in the title of the work.

b)      The Abstract does not briefly explain the methodology used, nor are quantitative results presented.

c)      The motivation presented is incomplete, as the use of one of the cell phone's cameras to achieve information about the location / environment assumes that the user is cooperative with the task, but if the user is using the cell phone for another task, the images acquired by the back camera of the cell phone, could be black images or images of something irrelevant to the initial task.

d)     The state of the art presented in chapter 1 is too compact, as some of the work is summarized in 2 lines. On the other hand, the results produced in each of the works mentioned in the state of the art are not described.

e)      It is not clear whether figure 1 was created by the authors of this work or whether it is a figure already published in the literature and in the latter case, copyright may exist.

f)       At the end of chapter 1, the authors present their contributions to this article. But the contribution (a) and (b) appear to be similar or equivalent; and it is not clear what are the new features of this work when compared to works already published in the literature.

g)      Chapter 2 describes the construction of the dataset, which is normally written after the methodology (chapter 3).

h)      The results produced by the authors include the F1-score metric, but when making a comparison with works by other authors, this metric is omitted. Given that the F1-score is more realistic than “accuracy”, this metric should be included in table 5.

i)       The argument presented in section 4.5 is valid if the cell phone is not being used by the user. But as the objective of the work is to alert users when they are distracted using their cell phone, consequently the screens shown in figure 8 will not be displayed or, if they are active, they will certainly be in the background, and not showing the alerts as would be desired.

j)        The conclusions presented are too compact. There must be a paragraph explaining the scientific methodology used or implemented. Then the authors can briefly explain the construction of the dataset. After that, the main results should be presented. Finally, the authors should describe the new aspects of the methodology, that is, what distinguishes this methodology from others already published in the literature.

Round 2

Reviewer 1 Report

Comments and Suggestions for Authors

All my concerns were adequately addressed by the authors.

The article is now suitable for publication.

Reviewer 2 Report

Comments and Suggestions for Authors

The author has answered my main questions. However, the author's response to the issue of privacy protection is not convincing enough. I think it is necessary to add a discussion of the limitations of the algorithm in this article on privacy protection in the conclusion.

Comments on the Quality of English Language

Minor editing of English language required
